# Synthesis and Characterization of Andrographolide Derivatives as Regulators of βAPP Processing in Human Cells

**DOI:** 10.3390/molecules26247660

**Published:** 2021-12-17

**Authors:** Arpita Dey, Ran Chen, Feng Li, Subhamita Maitra, Jean-Francois Hernandez, Guo-Chun Zhou, Bruno Vincent

**Affiliations:** 1Institute of Molecular Biosciences, Mahidol University, Nakhon Pathom 73170, Thailand; atipra.4evr@gmail.com (A.D.); msubhamita4u@yahoo.com (S.M.); 2School of Pharmaceutical Sciences, Nanjing Tech University, Nanjing 211816, China; 202162118026@njtech.edu.cn (R.C.); fengli9203@163.com (F.L.); 3Institut des Biomolécules Max Mousseron, UMR5247 CNRS/Université de Montpellier/ENSCM, Faculté de Pharmacie, CEDEX 5, 34093 Montpellier, France; jean-francois.hernandez@umontpellier.fr; 4Centre National de la Recherche Scientifique, 2 rue Michel Ange, 75016 Paris, France

**Keywords:** Alzheimer’s disease, βAPP, andrographolide, α-secretase, β-secretase, neuroprotection

## Abstract

Alzheimer’s disease (AD) is a devastating neurodegenerative disorder, one of the main characteristics of which is the abnormal accumulation of amyloid peptide (Aβ) in the brain. Whereas β-secretase supports Aβ formation along the amyloidogenic processing of the β-amyloid precursor protein (βAPP), α-secretase counterbalances this pathway by both preventing Aβ production and triggering the release of the neuroprotective sAPPα metabolite. Therefore, stimulating α-secretase and/or inhibiting β-secretase can be considered a promising anti-AD therapeutic track. In this context, we tested andrographolide, a labdane diterpene derived from the plant *Andrographis paniculata*, as well as 24 synthesized derivatives, for their ability to induce sAPPα production in cultured SH-SY5Y human neuroblastoma cells. Following several rounds of screening, we identified three hits that were subjected to full characterization. Interestingly, andrographolide (8,17-olefinic) and its close derivative 14α-(5′,7′-dichloro-8′-quinolyloxy)-3,19-acetonylidene (compound **9**) behave as moderate α-secretase activators, while 14α-(2′-methyl-5′,7′-dichloro-8′-quinolyloxy)-8,9-olefinic compounds **31** (3,19-acetonylidene) and **37** (3,19-diol), whose two structures are quite similar although distant from that of andrographolide and **9**, stand as β-secretase inhibitors. Importantly, these results were confirmed in human HEK293 cells and these compounds do not trigger toxicity in either cell line. Altogether, these findings may represent an encouraging starting point for the future development of andrographolide-based compounds aimed at both activating α-secretase and inhibiting β-secretase that could prove useful in our quest for the therapeutic treatment of AD.

## 1. Introduction

Alzheimer’s disease (AD) is the most prevalent neurodegenerative disorder and the main cause of dementia worldwide. It is characterized by a progressive loss of memory and cognitive function, which ultimately lead to dementia and death. Pathologically, there is an accumulation of extracellular β-amyloid peptide (Aβ) in senile plaques and of intracellular hyper-phosphorylated tau-containing neurofibrillary tangles (NTFs) in the hippocampus and cerebral cortex. This is accompanied by a large panel of molecular events related to the progression of the disease including oxidative stress, neuroinflammation, mitochondrial dysfunction, altered calcium homeostasis, and apoptosis [1].

The Aβ peptides are produced from the β-amyloid precursor protein (βAPP) along the amyloidogenic pathway through the sequential cleavages by β-secretase (BACE1) and the heterotetrameric γ-secretase complex that also gives rise to the production of the sAPPβ, C99, and AICD metabolites [2]. At the same time, a major alternative non-amyloidogenic route involves α-secretase activity. It not only hampers the production of the amyloid peptide since cleavage occurs in the middle of the Aβ sequence, but it also leads to the secretion of the metabolite sAPPα with neuroprotective and neurotrophic powers [3]. Therefore, α-secretase activation appears as a promising therapeutic strategy aimed at preventing AD [4,5]. Unfortunately, sustained efforts to inhibit and/or modulate β- and γ-secretases or to activate α-secretases are still unsuccessful, this being mainly explained by the fact that these enzymes cleave numerous other substrates involved in vital physiological functions [6,7,8]. Hence, the use of plant-derived active compounds has increasingly been considered in recent years as an alternative to pharmacotherapies [9] because side-effects for most medicinally used natural products are considerably low or inexistent.

Andrographolide (Figure 1) is a bicyclic diterpenoid lactone present in the stem and leaves of *Andrographis paniculata*. As the major bioactive constituent of this traditional Asian medicinal herb, it supports its antioxidant, anti-microbial, anti-inflammatory, and anti-cancer properties [10]. Interestingly, several beneficial effects of andrographolide on physiological functions of the central nervous system have been reported in recent years as shown by the ability of the compound to stimulate adult neurogenesis in the mouse hippocampus [11] and the capacity of andrographolide analogues to promote neurite outgrowth in rat PC-12 cells [12]. Furthermore, several studies have very recently evidenced some positive effects of andrographolide on AD pathology. Firstly, it can strongly attenuate Aβ-induced microglial activation [13,14] and autophagy-associated cell death [15] in vitro. Secondly and most importantly, andrographolide administration has been shown to alleviate AD-associated phenotypes, including cognitive deficits, observed in both transgenic [16,17,18] and non-transgenic [19,20] models of the disease.

Mechanistically, it has been shown that andrographolide activates the canonical Wnt signaling pathway via an inhibition of GSK-3β in primary neurons [21] and in the aged rodent *Octodon degus* [22], and that it increases glucose uptake and utilization in a Wnt-dependent manner in the J20 transgenic mouse model of AD [23]. Interestingly, there exists a close relationship between Wnt loss of function and AD-associated neurodegeneration [24] mostly because of the ability of Wnt to repress the transcription of the β-secretase BACE1 [25]. Indeed, it has been established that andrographolide can shift the metabolism of βAPP towards the non-amyloidogenic pathway as shown by an increased production of C83 and a concomitant decrease in C99 and Aβ42/Aβ40 ratio in epithelial cells overexpressing βAPP [26].

Because its simplistic structural nature brings amenability for semi-synthetic modifications, andrographolide has given rise to many derivatives with potent therapeutic effects in diverse fields [27]. In this context, we have undertaken to design, synthesize, and test twenty-four andrographolide derivatives, classified as two series, for their ability to favorably modulate βAPP processing through the stimulation of α-secretase and/or the inhibition of β-secretase catalytic activities/expression in human neuroblastoma SH-SY5Y cells. A first screening allowed us to identify andrographolide as well as the three derivatives **9**, **31,** and **37** as potent sAPPα production-enhancing compounds without altering cell viability. Their subsequent full characterization indicated that while andrographolide and **9** could moderately stimulate the α-secretase catalytic activity, **31** and **37** behaved as potent β-secretase inhibitors. These results thus established andrographolide and some of its derivatives as a promising basis for the future development of anti-amyloidogenic factors, the next generation of which hopefully leading to the setup of druggable α-secretase activator/β-secretase inhibitor compounds.

## 2. Results

### 2.1. Synthesis of Andrographolide Derivatives

Since andrographolide can exert neuroprotective effects [26,28,29] and because the quinoline moiety is important in anti-AD compounds, thanks to its antioxidant, anti-aggregating, and neurotrophic properties [28,30,31], we were interested in assessing the anti-AD activity of 24 andrographolide derivatives bearing a 14-quinolinyloxy group, and divided in 12 derivatives [32,33] of a 8,17-olefinic series (Figure 1) and 12 derivatives of a 9-dehydro-17-hydro series (Figure 2).

8,17-Olefinic compounds of **7** to **12** and their corresponding deprotected derivatives **13** to **18** were synthesized as previously published [32,33,34]. Briefly, the acetonide-protected 14α and 14β analogues of andrographolide [34] were reacted with various 8-hydroxyquinoline derivatives in Mitsunobu conditions, yielding the analogues **7**–**12**, which were subsequently deprotected by hydrolysis, giving the analogues **13**–**17** (Figure 1).

The analogues of 17-hydro-9-dehydro andrographolide (**27**–**32** and **33**–**38**) were prepared as presented in Figure 2. Firstly, 3,19-acetonylidene andrographolide (**2**) was 4-nitrobenzoylated by acylation or Mitsunobu reaction to form **19** or **20**, respectively. The key step is to isomerize the 8,17-double bond of **19** or **20** into 8,9-double bond of **21** or **22** by 85% H_3_PO_4_, which removed the acetonide protection. After re-protection as 3,19-acetonylidene, the 4-nitrobenzoyl group was removed to form the two key intermediates **25** (14α) and **26** (14β), which then reacted with the 8-hydroxyquinoline derivatives **4**, **5,** and **6** to yield the 3,19-acetonylidene-protected analogues **27** to **32**. Deprotection of the 3,19-acetonylidene group afforded the 3,19-diol analogues **33** to **38**.

### 2.2. Screening of Andrographolide Derivatives for sAPPα Production and sAPPα/βAPP Ratio

We then investigated the effect of andrographolide and its 24 derivatives for their ability to promote the secretion of the βAPP-derived sAPPα metabolite in cultured naive SH-SY5Y human neuroblastoma cells. As a first step, we chose to treat the cells for 24 h with 1 μM of the compounds (andrographolide, the 12 derivatives of the 8,17-olefinic series (8,17-double bond) and the 12 derivatives of the 9-dehydro-17-hydro series (8,9-olefen/double bond)) and we used both sAPPα production and the sAPPα/βAPP/β-actin ratio as a read out for comparison with controls (duplicate).

The results showed that four derivatives (**28**, **31**, and their corresponding deprotected forms **34** and **37**) of the 9-dehydro-17-hydro series were able to increase sAPPα production by a factor greater than two when compared to controls (Figure 1A,B), while treatment of cells with **9** and **31** led to a 2.5-fold augmentation of the sAPPα/βAPP/β-actin ratio (Figure 1C). It is noted that most of the tested andrographolide analogues were more active than andrographolide in increasing sAPPα production (Figure 1A,B) and sAPPα/βAPP ratio (Figure 1C). Specifically, eight compounds (**8**, **11**–**16**, and **18**) of the 8,17-double bond series and ten compounds (**28**–**34** and **36**–**38**) of the 8,9-double bond series increased sAPPα production when compared to andrographolide (Figure 1A,B), while nine compounds (**8**, **9**, **11**, **12**–**16**, and **18**) of the 8,17-double bond series and eight (**28**, **30**, **31**, **33**, **34**, and **36**–**38**) of the 8,9-double bond series displayed higher sAPPα/βAPP ratio than andrographolide (Figure 1C). Particularly, the 14α-(2′-methyl-5′,7′-dichloro-8′-quinolyloxy)-3,19-acetonylidene-8,17-olefinic analogue **31** showed the highest sAPPα secretion (Figure 1A,B) and the second highest sAPPα/βAPP ratio (Figure 1C). In contrast, 14α-(5′,7′-dichloro-8′-quinolyloxy)-8,9-olefin-3,19-diol **35** exhibited the lowest sAPPα secretion rate (Figure 1A,B) and sAPPα/βAPP ratio (Figure 1C).

Moreover, both 14α-(2′-methyl-8′-quinolyloxy)-3,19-acetonylidene compound **7** and its 8,9-double bond counterpart **27** displayed much lower sAPPα production (Figure 1A,B) and sAPPα/βAPP ratio (Figure 1C) than their corresponding 14β compounds **8** and **28**. Meanwhile, the 3,19-diols **13** (14α) and **14** (14β) are more able than the 3,19-protected compounds **7** and **8** to increase sAPPα production (Figure 1A,B) and sAPPα/βAPP ratio (Figure 1C) while the 14β-8,9-olefinic-3,19-diol **34** is superior to its 14α counterpart **33** in augmenting both sAPPα production (Figure 1A,B) and the sAPPα/βAPP ratio (Figure 1C).

Among the 5′,7′-dichloro-8′-quinolyloxy series, the three 8,17-olefinic compounds, 14α-3,19-acetonylidene **9**, 3,19-diols **15** (14α) and **16** (14β) are similarly active and superior to their 14β-3,19-acetonylidene-8,17-olefinic **10**, 14α-3,19-acetonylidene-8,9-double bond **29** and 14α-3,19-acetonylidene-8,9-olefinic **35** counterparts in both promoting sAPPα secretion (Figure 1A,B) and increasing the sAPPα/βAPP ratio (Figure 1C). In addition, it appeared that the 14β-3,19-acetonylidene-8,9-olefinic compound **30** is less active than its corresponding diol **36** in promoting sAPPα secretion (Figure 1A,B) and increasing the sAPPα/βAPP ratio (Figure 1C). In this series, **36** is the most active sAPPα secretion inducer (Figure 1A,B) while **9**, **16**, and **36** are the compounds most capable of increasing the sAPPα/βAPP ratio (Figure 1C).

Now regarding the 14-(2′-methyl-5′,7′-dichloro-8′-quinolyloxy)-8,17-olefinic analogues, the 14α-3,19-acetonylidene compound **11** and its 3,19-diol analogue **17** were less active than their respective 14β **12** and **18** counterparts in both sAPPα secretion (Figure 1A,B) and sAPPα/βAPP ratio (Figure 1C). On the other hand, the 14α-(2′-methyl-5′,7′-dichloro-8′-quinolyloxy)-8,9-olefinic compound **31** and its 3,19-diol analogue **37** displayed the highest sAPPα secretion rate (Figure 1A,B) and a relatively high sAPPα/βAPP ratio (Figure 1C). However, the corresponding 14β-3,19-acetonylidene compound **32** and its diol **38** behaved in opposite ways with **32** having a much lower ability than **38** to increase sAPPα secretion (Figure 1A,B) and the sAPPα/βAPP ratio (Figure 1C).

Overall, these results suggest that an optimal combination of 8,9-double bond or 8,17-double bond, substitution at quinolone, 14α or 14β, and 3,19-free diol or protection will benefit the enhancement of sAPPα production. As a whole, based on the screening results and structure-activity consideration, we undertook to focus on **9**, **28**, **31**, **34**, and **37** for further characterization, using andrographolide as the reference compound.

### 2.3. Further Characterization of Andrographolide Derivatives ***9***, ***28***, ***34***, ***31***, and ***37***

The results obtained following a consistent number of independent experiments (n ≥ 6 when compared with n = 2 for the initial screening step) first showed that **31** and **37** (1 μM) significantly induce sAPPα secretion (Figure 2A,B). Secondly, none of the compounds significantly altered βAPP immunoreactivity (Figure 2A,C), thereby ruling out an effect on βAPP expression or maturation and rather suggesting that they genuinely control βAPP processing. The additional measurement of the sAPPα/βAPP/β-actin ratio further indicated that all the selected derivatives including andrographolide itself were able to significantly increase this ratio although to different degrees (Figure 2D).

### 2.4. Full Characterization of Andrographolide Derivatives ***9***, ***31***, and ***37***

#### 2.4.1. Effect of Derivatives **9**, **31**, and **37** on Cell Survival

In light of these results, we decided to reduce our field of investigation to **9**, **31**, and **37**. At this stage, it was important to demonstrate that these compounds are not inherently toxic. For this purpose, we measured the survival rate of SH-SY5Y cells following a 24 h treatment at concentrations ranging from 100 nM to 10 μM with the MTT assay. In fact, no notable changes were observed whatever the compound and the concentrations considered (Figure 3), clearly indicating that none of them are toxic under our experimental conditions.

#### 2.4.2. Dose-Dependent Effect of Derivatives **9**, **31**, and **37** on sAPPα Production, βAPP Protein Levels, and sAPPα/βAPP Ratio

Following the screening phase carried out at one single concentration (1 μM), we then examined the ability of the selected compounds to stimulate sAPPα at lower concentrations and in a dose-dependent manner in SH-SY5Y cells. Although an increasing trend was observed for both andrographolide and **9**, we could not establish statistically significant differences with controls (Figure 4A). Nevertheless, in addition to the fact that **31** and **37** were triggering a significant increase in sAPPα production at 1 μM (Figure 4A) as previously observed (see Figure 2), we showed that **37** is also effective at 10 nM and 100 nM concentrations (Figure 4A). Parallel analysis of βAPP protein levels showed no significant variation in βAPP immunoreactivity under any conditions (Figure 4B). The concomitant measurement of the sAPPα/βAPP/β-actin ratio showed in addition that the four compounds increased it significantly at the highest concentrations (100 nM to 1 μM) (Figure 4C). Following the demonstration of the superior efficiency of compound **37** in inducing the production of sAPPα and in order to show that these effects are not restricted to a cell type but rather represent a ubiquitous phenomenon, we conducted the same experiments in human cells HEK293. The results showed that **37** produced effects similar and even superior to those observed in SH-SY5Y cells, namely an increase in sAPPα production and in the sAPPα/βAPP/β-actin ratio at all concentrations tested (Figure 4D, upper and lower panels, respectively). The additional observation that **37** also significantly reduced βAPP protein levels at the same concentrations in HEK293 cells (Figure 4D, middle panel) most likely reflects some depletion of the substrate due to higher metabolic activity when compared to the SH-SY5Y cell line.

#### 2.4.3. Effect of Derivatives **9**, **31**, and **37** on ADAM10 and BACE1 Protein and mRNA Levels

Based on these results, we then wanted to determine whether these compounds were capable of influencing the expression of the main α-secretase activity ADAM10 and of the β-secretase BACE1.

Firstly, the Western blot analyses of ADAM10 (Figure 5A) and BACE1 (Figure 5B) in SH-SY5Y cells did not detect any significant changes between the control conditions and those where the cells were treated with the four compounds at concentrations ranging from 1 nM to 1 μM. These results were then confirmed for **37** in HEK293 cells (Figure 5C). Because protein level measurement results from transcriptional, translational, and post-translational events, we undertook to examine the genuine transcriptional effect of the four compounds (1 μM) by real time qPCR in both SH-SY5Y and HEK293 cells. The results indicated a slight but significant increase in ADAM10 mRNA levels when cells were treated with 1 μM of **31** and **37** (SH-SY5Y) or **9** (HEK293) (Figure 5D), while no significant change in BACE1 mRNA levels was detected (Figure 5E).

This set of data suggests transcriptional up-regulation of the α-secretase ADAM10 as a minor although possibly involved mechanism in the observed beneficial effect of these compounds on βAPP metabolism.

We finally subjected andrographolide and the three derivatives to a thorough characterization aimed at evaluating their effect on the catalytic activities of α- and β-secretases, which compete for βAPP processing, thereby tightly controlling the balancing between the amyloidogenic and the non-amyloidogenic pathways.

#### 2.4.4. Effect of Derivatives **9**, **31**, and **37** on α-Secretase Catalytic Activity

In a first set of experiments, we examined the impact of increasing concentrations (10 nM up to 10 μM for andrographolide and 1 nM to 1 μM for compounds **9**, **31**, and **37**) of the four compounds on the α-secretase activity by measuring the phenanthroline-sensitive hydrolysis of the fluorimetric JMV2770 substrate by cultured SH-SY5Y cells.

Our results indicated that andrographolide slightly and dose-dependently enhances the JMV2770-hydrolyzing activity, **9** displaying such capability only at 10 nM while **31** and **37** remain inert in this paradigm (Figure 6A). Andrographolide and **9** were subsequently submitted to the same assay in HEK293 where they also significantly contributed to a moderate stimulation of the α-secretase activity, although showing a slightly different pattern when compared to SH-SY5Y cells (Figure 6B).

#### 2.4.5. Effect of Derivatives **9**, **31**, and **37** on β-Secretase Catalytic Activity

Another important aspect of this study was to determine if these compounds could behave as inhibitors of the amyloidogenic β-secretase catalytic activity. Taking advantage of a well-characterized BACE1-selective fluorimetric assay, we have first measured the effect of the four molecules, at the same concentrations used for the α-secretase assay, on the JMV1197-sensitive hydrolysis of the fluorimetric JMV2236 substrate in SH-SY5Y cell extracts at acidic pH. We showed a capability of all the compounds to reduce BACE1 activity, **31** and **37** operating the most efficiently and in a dose-dependent manner (Figure 7A).

The data obtained with **31** and **37** were then reproduced with HEK293 cell extracts, thereby confirming the genuine ability of these two andrographolide analogues to potently block the β-secretase catalytic activity (Figure 7B). It should be noted here that the inhibitory effects of the compounds on the β-secretase activity seem to be more pronounced than their capability to stimulate the α-secretase activity. From the fact that compound **9** of the 8,17-olefinic series possesses anti-BACE1 activity, it is suggested that the 5′,7′-dichloro-8′-hydroxyquinolyloxy moiety is important for BACE1 inhibition. As the 2′-methyl-5′,7′-dichloro-8′-hydroxyquinolyloxy derivatives **31** and **37** are potent BACE1 inhibitors, this particular structure might interact in a more efficient way with the BACE1 catalytic site. Finally, the observation that **31** and **37** are the most potent β-secretase inhibiting factors in this study, suggests that the 8,9-double bond in the 9-dehydro-17-hydro series is an important feature for a proper inhibition of this activity.

## 3. Discussion

AD is a yet incurable neurodegenerative disorder characterized by loss of memory and cognition. The reason why available medical treatments are still incapable of curing AD symptoms efficiently mostly resides in the fact that AD is a complex and multifactorial pathology. Over the past decades, a huge effort, although in vain, has been made to develop novel synthetic drugs with disease-modifying properties and few side effects [35]. Hence, compounds extracted from natural sources are constantly gaining popularity in AD treatment with the notion of preventive rather than curative intervention against the disease being increasingly considered.

In this context, beside its previously reported effects on viral infection [36], bacterial infection [37], cancer [38], metabolic syndromes [39], and inflammation [40], thereby making this molecule a multi-targeting agent [41], andrographolide has also been interestingly established as a promising candidate in neuropharmacology as it shows diverse potent therapeutic effects against various neurological disorders [29], such as brain ischemic stroke [42], multiple sclerosis [43], Parkinson’s disease [44], and Alzheimer’s disease [19].

Regarding AD and on a mechanistic point of view, andrographolide most likely conveys some anti-AD effects via its well-established antioxidant [45] and NFκB inhibitory and anti-inflammatory [46] properties as illustrated for instance by the fact that andrographolide inhibits Aβ_1–42_-induced production of neuroinflammatory mediators in microglia [13,14]. However, whether it could regulate the processing of βAPP through the control of βAPP-cleaving secretases was still an unanswered question. Here, we first identified andrographolide as well as some chemically modified andrographolide analogues as regulators of βAPP processing in cultured human cell lines using sAPPα production as a read-out. This metabolite with beneficial properties arises from the cleavage of βAPP by the non-amyloidogenic α-secretase activity. Because α-secretase and β-secretase, the amyloidogenic rate-limiting initiator of amyloid peptide production, work competitively regarding βAPP processing as evidenced by the inverse correlation between sAPPα and Aβ productions under both α-secretase activation or β-secretase inhibition [47,48], any increase in sAPPα production can result from either an activation of α-secretase or an inhibition of β-secretase (that disrupts sAPPα integrity by cleaving inside its sequence), or both. We therefore undertook to study the effect of andrographolide, **9**, **31**, and **37**, all initially identified on the basis of their ability to induce the production of sAPPα, on the catalytic activities of α- and β-secretase by means of specific fluorimetric assays. This allowed us to establish that the four compounds indeed regulate these activities although to different degrees and that **31** and **37** behave as potent β-secretase inhibitors. It has to be underlined here that their efficiency at submicromolar concentrations lays the groundwork for the future production of highly potent derivatives that could serve as a basis for their therapeutic use. Moreover, the confirmation of the results in HEK293 cells suggests that the effects observed are probably ubiquitous and not restricted to one cell type.

Importantly, studies carried out in animals have shown that andrographolide does not trigger toxicity in the liver and the kidney of rat [49] and does not alter body and organ weight, inflammatory responses, hematological parameters, and mortality in mice [50]. These data therefore established andrographolide as a relatively safe compound in respect to toxicological side effects. Moreover, andrographolide easily passes the blood–brain barrier and distributes into different brain regions [51]. However, restricted bioavailability due to its poor solubility and relatively short half-life obviously limits its clinical application and numerous semi-synthetic transformations were performed in order to improve its physiochemical properties and stability [52,53].

Considering that quinoline is also a pharmacophore group for neuroprotection [28,30,31], we envisaged that our published active 14-quinolyloxy derivatives of andrographolide against Zika and dengue viruses [32,33] and bacteria [54] possibly have anti-AD activity. Moreover, in addition to andrographolide itself [26,29,36,55], some of its 9-dehydro-17-hydro analogues have increased neuroprotective properties [12] and are more efficient against angiogenesis [56,57] than its 8,17-olefinic counterpart. These results led us to explore whether 14-aryloxy-9-dehydro-17-hydro analogues also possess a higher capability to inhibit BACE1 activity than the 8,17-olefinic ones. Our results confirmed the concept that 14-quinolyloxy modification or combination of 14-quinolyloxy and 9-dehydro-17-hydro modifications on andrographolide benefits for neuroprotection and will lead to the discovery of more potent and druggable anti-AD compounds.

Overall, our original findings that andrographolide derivatives display both pro-α-secretase and anti-β-secretase properties open the way to the possible development, hitherto not explored, of molecules of natural origin capable of acting in a doubly beneficial manner on the metabolism of βAPP and representing a new class of factors to be developed as a therapeutic tool aimed at combating Alzheimer’s disease. The design, synthesis and testing of new chemically modified andrographolide-derived compounds aimed at obtaining highly potent α- and β-secretases regulating molecules is currently being carried out in our laboratories.

## 4. Materials and Methods

### 4.1. Materials

DMEM, fetal bovine serum (FBS), and penicillin-streptomycin mix (Pen/Strep) were from Invitrogen (Carlsbad, CA, USA). Tris buffer and glycine were from VWR Amresco lifesciences (Solon, CA, USA). Polyclonal anti-βAPP antibody (A8717), monoclonal anti-β-actin (A2228), dimethyl sulfoxide (DMSO), SDS, and sodium bicarbonate were from Sigma (St. Louis, MO, USA). Polyclonal anti-ADAM10 (AB19026) was from Millipore (Bedford, MA, USA). Monoclonal anti-BACE1 (ab108394) was from Abcam (Cambridge, UK). Skim milk powder was from Bio Basic (Singapore). Monoclonal anti-β-amyloid antibody (2B3), which was used to specifically detect sAPPα was from IBL (Minneapolis, MN, USA). ECL reagent and ammonium persulphate were from GE Health care (Pisataway, NJ, USA). *O*-Phenanthroline was from Calbiochem (San Diego, CA, USA). Goat anti-mouse (polyclonal 7076) and goat anti-rabbit (polyclonal 7074) peroxidase-conjugated secondary antibodies were from Cell Signaling (Beverly, MA, USA).

### 4.2. General Information for Chemistry

^1^H and ^13^C NMR spectra (Appendix A) were recorded on a Bruker AV-400 spectrometer at 400 and 100 MHz, respectively, in CDCl_3_, CD_3_OD, (CD_3_)_2_SO, and C_6_D_6_ as indicated. Coupling constants (*J*) are expressed in hertz (Hz). Chemical shifts (*δ*) of NMR are reported in parts per million (ppm) units relative to the solvent. The high resolution of ESI-MS was recorded on an Applied Biosystems Q-STAR Elite ESI-LC-MS/MS mass spectrometer, respectively. Unless otherwise noted, materials were obtained from commercial suppliers and used without further purification. Melting points were measured using an YRT-3 melting point apparatus (Shanghai Yice Apparatus & Equipment Co., Ltd., Shanghai, China) and were uncorrected.

### 4.3. Synthesis of Andrographolide Derivatives

#### 4.3.1. Synthesis of Compounds **7** to **10** and **12** to **16**

The synthesis of compounds **7** to **10** and **12** to **16** (Figure 1) was previously described [32,33].

#### 4.3.2. Synthesis of Compounds **11**, **17**, and **18**

The synthesis of compounds **11**, **17**, and **18** (Figure 1) was conducted according to previously described procedures [32,33,34].

(14α)-(Quinolyl-2′-methyl-5′,7′-dichloro-8′-oxy)-3,19-acetonylidene andrographolide (**11**): white solid; m.p. 166.3–167.0 °C; 72.6% yield; ^1^H NMR (400 MHz, DMSO-*d*_6_) *δ* 8.48 (d, *J* = 8.7 Hz, 1H), 7.92 (s, 1H), 7.69 (d, *J* = 8.7 Hz, 1H), 6.74–6.65 (m, 1H), 6.37 (d, *J* = 4.6 Hz, 1H), 4.75–4.68 (m, 2H), 4.60 (m, 1H), 4.10 (s, 1H), 3.76 (d, *J* = 11.6 Hz, 1H), 3.31–3.26 (m, 1H), 3.03 (d, *J* = 11.6 Hz, 1H), 2.77 (s, 3H), 2.23 (d, *J* = 12.2 Hz, 1H), 1.90–1.69 (m, 3H), 1.63 (d, *J* = 11.6 Hz, 1H), 1.55 (m, 3H), 1.31 (s, 3H), 1.24 (s, 3H), 1.09 (d, *J* = 4.4 Hz, 2H), 1.06 (s, 3H), 0.92 (m, 2H), 0.46 (s, 3H); ^13^C NMR (101 MHz, DMSO) *δ* 170.0, 160.9, 148.7, 147.7, 147.2, 142.9, 133.8, 127.4, 126.9, 126.6, 124.7, 124.5, 108.3, 98.6, 77.0, 76.2, 72.3, 63.1, 54.8, 51.8, 38.2, 37.5, 37.3, 34.2, 28.0, 26.2, 25.8, 25.5, 25.2, 25.1, 23.0, 15.6; HRMS (ESI) *m*/*z* 600.2280 [M + H]^+^, calculated for C_33_H_40_Cl_2_NO_5_, 600.2284.

(14α)-(Quinolyl-2′-methyl-5′,7′-dichloro-8′-oxy) andrographolide (**17**): white solid; m.p. 142.0–142.4 °C; 75.1% yield; ^1^H NMR (400 MHz, DMSO-*d*_6_) *δ* 8.48 (d, *J* = 8.7 Hz, 1H), 7.91 (s, 1H), 7.75–7.61 (m, 1H), 6.68 (dd, *J* = 8.7, 5.0 Hz, 1H), 6.35 (d, *J* = 4.6 Hz, 1H), 5.09–4.92 (m, 1H), 4.74–4.65 (m, 2H), 4.60 (m, 1H,), 4.07 (d, *J* = 19.2 Hz, 2H), 3.70 (d, *J* = 10.9 Hz, 1H), 3.16 (d, *J* = 10.8 Hz, 1H), 3.07 (m, 1H), 2.76 (s, 3H), 2.24–2.14 (m, 1H), 1.77 (m, 2H), 1.59 (m, 2H), 1.45–1.40 (m, 2H), 1.26–1.13 (m, 2H), 0.99 (m, 4H), 0.80 (m, 2H), 0.22 (s, 3H); ^13^C NMR (101 MHz, DMSO-*d*_6_) *δ* 169.6, 160.5, 148.2, 147.3, 146.8, 142.5, 133.5, 127.0, 126.5, 126.2, 126.1, 124.2, 124.1, 107.4, 78.2, 76.6, 71.9, 62.5, 54.4, 54.1, 42.2, 38.3, 37.2, 36.0, 27.7, 25.1, 24.4, 23.9, 23.0, 14.1; HRMS (ESI) *m*/*z* 560.1972 [M + H]^+^, calculated for C_30_H_36_Cl_2_NO_5_, 560.1971.

(14β)-(Quinolyl-2′-methyl-5′,7′-dichloro-8′-oxy) andrographolide (**18**): white solid; m.p. 172.1–172.7 °C; 74.4% yield; ^1^H MR (400 MHz, DMSO-*d*_6_) *δ* 8.46 (d, *J* = 8.7 Hz, 1H), 7.90 (s, 1H), 7.68 (d, *J* = 8.8 Hz, 1H), 6.59 (m, *J* = 8.5, 4.6 Hz, 1H), 6.48 (d, *J* = 4.5 Hz, 1H), 5.00 (d, *J* = 4.8 Hz, 1H), 4.72–4.64 (m, 2H), 4.60 (m, *J* = 11.0, 4.7 Hz, 1H), 4.13 (s, 1H), 4.07–4.04 (m, 1H), 3.71 (m, *J* = 10.9, 2.9 Hz, 1H), 3.15 (m, 1H), 3.09 (m, 1H), 2.75 (s, 3H), 2.23–2.13 (m, 1H), 1.83–1.68 (m, 2H), 1.59 (d, *J* = 12.3 Hz, 2H), 1.45 (q, *J* = 6.6 Hz, 2H), 1.36–1.20 (m, 2H), 1.05–0.92 (m, 4H, 1-H), 0.84–0.75 (m, 1H), 0.68 (m, 1H), 0.19 (s, 3H); ^13^C NMR (101 MHz, DMSO-*d*_6_) *δ* 169.5, 160.5, 149.2, 147.6, 146.7, 142.6, 133.3, 127.2, 126.5, 126.3, 125.9, 124.2, 124.1, 107.8, 78.2, 76.7, 71.8, 62.5, 54.4, 54.0, 42.1, 38.1, 37.2, 35.9, 27.8, 25.1, 25.0, 23.9, 23.0, 14.3; HRMS (ESI) *m*/*z* 560.1965 [M + H]^+^, calculated for C_30_H_36_Cl_2_NO_5_, 560.1971.

#### 4.3.3. Synthesis of Compounds **19** to **26**

The key intermediates **25** and **26** of the 9-dehydro-17-hydro series were synthesized as shown in Figure 2.

Synthesis of compound **19**: a solution of compound **2** (5.0 g, 12.8 mmol) in 10 mL of dry dichloromethane (50 mL) was cooled in ice-water bath and then triethylamine (4.5 mL, 32.0 mmol) was added, followed by *p*-nitrobenzoyl chloride (2.85 g, 15.4 mmol) in 20 mL of dry dichloromethane. The reaction mixture was stirred in an ice-water bath for 5 h and volatile solvents were removed by Rotavapor. The residue was dissolved in ethyl acetate and treated with sat. NaHCO_3_ aqueous solution. The organic phase was washed with brine twice and then dried over anhydrous Na_2_SO_4_. The filtered organic solution was evaporated to dryness and the residue was purified by silica gel column chromatography (petroleum ether/ethyl acetate 3/1) to afford 5.9 g of titled compound **19**. (14α)-(4′-nitrobenzoyl)-3,19-isopropylideneoxy andrographolide (**19**): white solid; m.p. 82.7–84.1 °C; 86.9% yield; ^1^H NMR (400 MHz, Chloroform-*d*) *δ* 8.37–8.28 (m, 2H), 8.25–8.14 (m, 2H), 7.14 (m, 1H), 6.24 (d, *J* = 5.9 Hz, 1H), 4.85 (s, 1H), 4.68 (m, 1H), 4.50 (s, 1H), 4.41 (m, 1H), 3.91 (d, *J* = 11.5 Hz, 1H), 3.48 (m, 1H), 3.15 (d, *J* = 11.6 Hz, 1H), 2.54–2.47 (m, 1H), 2.43–2.36 (m, 1H), 2.02–1.92 (m, 2H), 1.89 (d, *J* = 10.0 Hz, 1H), 1.81–1.74 (m, 1H), 1.69 (m, 2H), 1.37 (s, 3H), 1.35 (s, 3H), 1.30 (t, *J* = 6.4 Hz, 1H), 1.26 (d, *J* = 3.7 Hz, 2H), 1.18 (s, 3H), 0.87 (s, 3H), 0.86–0.85 (m, 1H).

Synthesis of compound **20**: to a solution of compound **2** (5.0 g, 12.8 mmol), *p*-nitrobenzoic acid (2.57 g, 15.4 mmol) and triphenylphosphine (5.0 g, 19.2 mmol) in anhydrous THF (50 mL) placed under N_2_ and at 0 °C, diisopropyl azodicarboxylate (DIAD) (3.76 mL, 19.2 mmol) was added. The reaction was stirred at 0 °C for 1 h and at room temperature overnight. After the reaction was complete as established by TLC monitoring, the volatile solvents were distilled off, the residue was dissolved in ethyl acetate and the organic phase was washed with brine twice, dried over anhydrous Na_2_SO_4_, filtered, and evaporated to dryness. Titled compound **20** was purified by silica gel column chromatography (petroleum ether/ethyl acetate 8/1) to yield 5.3 g. (14β)-(4′-nitrobenzoyl)-3,19-isopropylideneoxy andrographolide (**20**): white solid; m.p. 109.6–110.8 °C; 77.5% yield; ^1^H NMR (400 MHz, Chloroform-*d*) *δ* 8.36 (d, *J* = 8.9 Hz, 2H), 8.24 (d, *J* = 8.8 Hz, 2H), 7.21–7.15 (m, 1H), 6.29 (d, *J* = 5.8 Hz, 1H), 5.00 (m, 1H), 4.93 (s, 1H), 4.71 (m, 1H), 4.49 (s, 1H), 4.43 (m, 1H), 3.88 (d, *J* = 11.6 Hz, 1H), 3.41–3.28 (m, 1H), 3.14 (d, *J* = 11.6 Hz, 1H), 2.54–2.50 (m, 1H), 2.44 (d, *J* = 13.3 Hz, 1H), 2.01–1.87 (m, 2H), 1.71 (m, 2H), 1.36 (s, 3H), 1.34 (s, 3H), 1.30 (s, 4H), 1.16 (d, *J* = 2.6 Hz, 1H), 1.11 (s, 3H), 0.92 (s, 3H).

Synthesis of compound **21**: compound **19** (5.9 g, 10.9 mmol) was added to 85% phosphoric acid (40.0 mmol) with fast stirring and the solid was dissolved gradually. The reaction was monitored by TLC and complete in about 3 h before being diluted carefully with distilled water followed by extraction with ethyl acetate. The organic phase was washed with a sat. NaHCO_3_ aqueous solution and brine and then dried over anhydrous Na_2_SO_4_. Filtered organic phase was evaporated and the residue was purified by silica gel column chromatography (petroleum ether/ethyl acetate 3/1) to yield 3.6 g of compound **21**. (14α)-(4′-nitrobenzoyl)-9-dehydro-17-hydro andrographolide (**21**): white solid; m.p. 97.3–98.5 °C; 66.7% yield; ^1^H NMR (400 MHz, Chloroform-*d*) *δ* 8.39–8.27 (m, 2H), 8.23–8.14 (m, 2H), 6.99 (m, 1H), 6.26 (d, *J* = 5.8 Hz, 1H), 4.68 (m, 1H), 4.42 (m, 1H), 4.17 (d, *J* = 11.2 Hz, 1H), 3.40 (d, *J* = 8.8 Hz, 1H), 3.30 (s, 1H), 3.08 (m, 1H), 2.95 (m, 1H), 2.73 (d, *J* = 8.5 Hz, 1H), 2.53 (s, 1H), 2.03 (m, 2H), 1.82–1.64 (m, 4H), 1.50 (s, 3H), 1.40–1.27 (m, 2H), 1.23 (s, 3H), 1.20–1.16 (m, 1H), 0.89 (s, 3H). ^13^C NMR (101 MHz, DMSO-*d*_6_) δ 169.4, 164.3, 150.9, 150.2, 136.7, 134.9, 131.4 (2C), 129.1, 124.4 (2C), 123.9, 78.6, 71.6, 69.8, 63.2, 51.7, 42.6, 38.6, 34.9, 34.4, 28.5, 28.0, 23.3, 20.5, 19.7, 19.4. HRMS (ESI) *m*/*z* 522.2101 [M + Na]^+^, calculated for C_27_H_33_NO_8_Na, 522.2098.

Synthesis of compound **22**: compound **22** was synthetized from compound **20** following the synthetic procedure described for the synthesis of compound **21**. (14β)-(4′-nitrobenzoyl)-9-dehydro-17-hydro andrographolide (**22**): white solid; m.p. 107.8–109.1 °C; 68.5% yield; ^1^H NMR (400 MHz, Chloroform-*d*) *δ* 8.33 (d, *J* = 8.9 Hz, 2H), 8.20 (d, *J* = 8.8 Hz, 2H), 6.99 (t, *J* = 7.0 Hz, 1H), 6.27 (d, *J* = 5.8 Hz, 1H), 4.68 (m, 1H), 4.42 (m, 1H), 4.15 (d, *J* = 11.2 Hz, 1H), 3.49–3.39 (m, 1H), 3.30 (d, *J* = 11.1 Hz, 1H), 3.09 (m, 1H), 2.96 (m, 1H), 2.66 (s, 1H), 2.43 (s, 1H), 2.03 (d, *J* = 8.0 Hz, 2H), 1.81–1.68 (m, 4H), 1.51 (s, 3H), 1.38–1.32 (m, 1H), 1.29 (d, *J* = 7.6 Hz, 1H), 1.24 (s, 3H), 0.90–0.86 (m, 1H), 0.85 (s, 3H). ^13^C NMR (101 MHz, Chloroform-*d*) δ 169.1, 164.2, 151.1, 151.0, 135.2, 134.1, 131.0, 130.4, 123.8, 122.8, 80.2, 71.5, 69.3, 64.1, 51.6, 42.7, 38.4, 35.0, 34.2, 28.8, 27.9, 22.5, 20.6, 19.5, 18.7. HRMS (ESI) *m*/*z* 522.2099 [M + Na]^+^, calculated for C_27_H_33_O_8_NNa, 522.2098.

Synthesis of compound **23**: compound **21** (3.6 g, 7.2 mmol) was dissolved in 2,2-dimethoxypropane (7.5 mL, 50.4 mmol) and 2.5 mL dry dichloromethane and pyridinium 4-toluenesulfonate (88 mg, 0.36 mmol) was added. The reaction was stirred at 45 °C and complete in 3 h as monitored by TLC. After volatile solvents were distilled off, the residue was taken off with ethyl acetate and the organic phase was washed with sat. CuSO_4_ aqueous solution, sat. NaHCO_3_ solution, and brine. The organic phase was dried over anhydrous Na_2_SO_4_, filtered, and evaporated in vacuo, and the residue was purified by silica gel column chromatography (petroleum ether/ethyl acetate 3/1) to give 3.1 g of compound **23**. (14α)-(4′-nitrobenzoyl)-9-dehydro-17-hydro-3,19-isopropylideneoxy andrographolide (**23**): white solid; m.p. 90.3–91.7 °C; 76.9% yield; ^1^H NMR (400 MHz, DMSO-*d*6) *δ* 8.40–8.33 (m, 2H), 8.24–8.17 (m, 2H), 6.78–6.72 (m, 1H), 6.32 (d, *J* = 5.8 Hz, 1H), 4.72 (m, 1H), 4.56 (m, 1H), 3.82 (d, *J* = 11.6 Hz, 1H), 3.29 (m, 1H), 3.15–3.00 (m, 3H), 1.97 (d, *J* = 7.3 Hz, 2H), 1.69–1.61 (m, 1H), 1.55 (s, 2H), 1.50 (s, 3H), 1.45 (d, *J* = 5.4 Hz, 1H), 1.37–1.27 (m, 1H), 1.21 (s, 6H), 1.14 (m, 2H), 1.06 (d, *J* = 1.8 Hz, 6H).

Synthesis of compound **24**: the procedure for the synthesis of compound **24** from compound **22** is the same as for the synthesis of compound **23**. (14β)-(4′-nitrobenzoyl)-9-dehydro-17-hydro-3,19-isopropylideneoxy andrographolide (**24**): white solid; m.p. 153.7-154.4 °C; 82.1% yield; ^1^H NMR (400 MHz, DMSO-*d*6) *δ* 8.90 (s, 2H), 8.43–8.32 (m, 2H), 8.27–8.17 (m, 2H), 6.74 (t, *J* = 6.7 Hz, 1H), 6.33 (d, *J* = 5.9 Hz, 1H), 4.80–4.70 (m, 3H), 4.57 (m, 1H), 3.83 (d, *J* = 11.6 Hz, 1H), 3.11 (d, *J* = 10.9 Hz, 1H), 1.50 (s, 3H), 1.40 (s, 2H), 1.23 (d, *J* = 3.2 Hz, 6H), 1.19 (s, 6H), 1.09 (s, 3H), 1.03 (s, 3H).

Synthesis of compound **25** (key intermediate): to a solution of compound **23** (3.1 g, 5.7 mmol) in 20 mL methanol, lithium carbonate (794 mg, 11.5 mmol) was added, and the mixture was stirred at room temperature for 2 h (TLC monitoring). After removal of volatile solvents by rotavapor, the residue was treated with ethyl acetate and the organic phase was washed with brine twice, dried over anhydrous Na_2_SO_4_, filtered, and distilled off to dryness. The residue was purified by silica gel column chromatography (petroleum ether/ethyl acetate 3/1) to give 1.6 g of titled compound **25**. (14α)-9-dehydro-17-hydro-3,19-isopropylideneoxy andrographolide (**25**): white solid; m.p. 138.2–139.4 °C; 71.4% yield; ^1^H NMR (400 MHz, Chloroform-*d*) δ 6.88 (m, 1H), 5.07 (s, 1H), 4.53–4.44 (m, 1H), 4.26 (m, 1H), 3.99 (d, *J* = 11.6 Hz, 1H), 3.48 (m, 1H), 3.26–3.14 (m, 2H), 3.04 (m, 1H), 2.04 (m, 2H), 1.99–1.92 (m, 1H), 1.78–1.69 (m, 2H), 1.68–1.61 (m, 2H), 1.55 (s, 3H), 1.50–1.43 (m, 1H), 1.41 (s, 3H), 1.37 (s, 3H), 1.32 (d, *J* = 6.2 Hz, 1H), 1.28 (d, *J* = 1.9 Hz, 1H), 1.25 (d, *J* = 2.1 Hz, 3H), 1.17 (s, 3H). ^13^C NMR (101 MHz, DMSO-*d*_6_) *δ* 170.4, 146.2, 137.6, 128.9, 128.1, 98.8, 75.7, 74.7, 65.2, 63.4, 48.6, 37.8, 37.6, 33.6, 32.2, 28.2, 27.3, 26.1, 25.8, 24.9, 22.2, 19.7, 18.32. HRMS (ESI) *m*/*z* 413.2299 [M + Na]^+^,calculated for C_23_H_34_O_5_Na, 413.2298.

Synthesis of compound **26** (key intermediate): the synthetic procedure for compound **26** from compound **24** is the same as that for compound **25**. (14β)-9-dehydro-17-hydro-3,19-isopropylideneoxy andrographolide (**26**): white solid; m.p. 93.2–94.7 °C; 72.6% yield; ^1^H NMR (400 MHz, Chloroform-*d*) δ 6.88 (t, *J* = 6.1 Hz, 1H), 5.10 (t, *J* = 6.5 Hz, 1H), 4.48 (m, 1H), 4.27 (m, 1H), 3.99 (d, *J* = 11.6 Hz, 1H), 3.50 (m, 1H), 3.21 (d, *J* = 11.6 Hz, 1H), 3.12 (d, *J* = 7.0 Hz, 2H), 2.07–1.96 (m, 3H), 1.93 (d, *J* = 6.7 Hz, 1H), 1.81–1.72 (m, 2H), 1.64 (m, 1H), 1.52 (s, 3H), 1.43 (s, 1H), 1.41 (s, 3H), 1.37 (s, 3H), 1.30 (d, *J* = 1.8 Hz, 1H), 1.28 (d, *J* = 5.8 Hz, 1H), 1.21 (s, 3H), 1.19 (s, 3H). ^13^C NMR (101 MHz, DMSO-*d*6) *δ* 170.4, 146.0, 137.6, 129.0, 128.3, 98.8, 75.9, 74.7, 65.2, 63.3, 48.8, 37.9, 37.5, 33.7, 32.6, 28.2, 27.5, 26.1, 25.9, 25.1, 22.1, 19.6, 18.3. HRMS (ESI) *m*/*z* 413.2297 [M + Na]^+^, calculated for C_23_H_34_O_5_Na, 413.2298.

#### 4.3.4. Synthesis of Compounds **27** to **32**

Titled 9-dehydro-17-hydro series compounds **27** to **32** were prepared as shown in Figure 2 according to previously described procedures [32,33,34]. Under N_2_ atmosphere, 1.0 mmol of compound **25** or **26**, 1.5 mmol of PPh_3_, and 1.5 mmol of 8-hydroxyquinoline derivative **4**, **5**, or **6** were dissolved in 10.0 mL of anhydrous THF. The solution was cooled to 0 °C and then treated with 1.5 mmol of DIAD in 2.0 mL of anhydrous THF. The reaction was stirred overnight at room temperature after being stirred at 0 °C for 1 h. After distilling off the volatile solvents, the residue was dissolved in ethyl acetate and the organic phase was washed with brine about 5 times and dried over anhydrous Na_2_SO_4_. The filtered organic solution was evaporated to dryness and the residue was purified by silica gel column chromatography to afford compounds **27** to **32**.

(14α)-(2′-Methyl-8′-quinolinoxy)-9-dehydro-17-hydro-3,19-isopropylideneoxy-andrographolide (**27**): white solid; m.p. 87.2–87.7 °C; 69.5% yield; ^1^H NMR (400 MHz, Chloroform-*d*) *δ* 8.08 (d, *J* = 8.5 Hz, 1H), 7.55 (d, *J* = 7.7 Hz, 1H), 7.42 (t, *J* = 7.8 Hz, 1H), 7.35 (d, *J* = 8.4 Hz, 1H), 7.16–7.13 (m, 1H), 7.00 (s, 1H), 6.17 (d, *J* = 5.6 Hz, 1H), 4.77 (m, 1H), 4.66 (m, 1H), 3.96 (d, *J* = 11.5 Hz, 1H), 3.46 (m, 1H), 3.19 (d, *J* = 11.6 Hz, 1H), 2.98 (d, *J* = 17.1 Hz, 1H), 2.78 (s, 3H), 2.09 (d, *J* = 45.1 Hz, 2H), 1.94–1.82 (m, 2H), 1.52 (s, 3H), 1.45 (s, 6H), 1.38 (s, 3H), 1.26 (s, 1H), 1.21 (d, *J* = 10.8 Hz, 2H), 1.16 (s, 3H), 1.12 (d, *J* = 6.2 Hz, 3H). HRMS (ESI) *m*/*z* 532.3057 [M + H]^+^, calculated for C_33_H_42_NO_5_, 532.3057.

(14β)-(2′-Methyl-8′-quinolinoxy)-9-dehydro-17-hydro-3,19-isopropylideneoxy-andrographolide (**28**): white solid; m.p. 99.6–100.3 °C; 76.2% yield; ^1^H NMR (400 MHz, Chloroform-*d*) *δ* 8.06 (d, *J* = 8.4 Hz, 1H), 7.53 (m, 1H), 7.39 (t, *J* = 7.9 Hz, 1H), 7.32 (d, *J* = 8.4 Hz, 1H), 7.13 (m, 1H), 6.97 (m, 1H), 6.14 (d, *J* = 5.8 Hz, 1H), 4.92 (d, *J* = 6.8 Hz, 1H), 4.74 (m, 1H), 4.62 (m, 1H), 3.95 (d, *J* = 11.6 Hz, 1H), 3.46 (m, 1H), 3.18 (d, *J* = 11.5 Hz, 1H), 2.89 (d, *J* = 6.8 Hz, 2H), 2.75 (s, 3H), 1.96 (m, 3H), 1.76–1.67 (m, 1H), 1.45 (s, 3H), 1.39 (s, 3H), 1.37 (s, 1H), 1.36 (s, 3H), 1.21 (m, 3H), 1.16 (s, 3H), 1.05 (s, 3H). ^13^C NMR (101 MHz, Chloroform-*d*) *δ* 170.2, 158.3, 152.0, 151.0, 141.2, 136.7, 136.4, 128.8, 128.2, 125.6, 124.9, 123.0, 122.6, 118.0, 99.2, 76.0, 74.9, 71.7, 63.9, 48.5, 37.7, 37.6, 33.6, 32.2, 28.9, 26.8, 25.9, 25.6, 25.5, 24.8, 22.1, 19.5, 18.3. HRMS (ESI) *m*/*z* 532.3060 [M + H]^+^, calculated for C_33_H_42_NO_5_, 532.3057.

(14α)-(5′,7′-Dichloro-8′-quinolinoxy)-9-dehydro-17-hydro-3,19-isopropylideneoxy-andrographolide (**29**): white solid; m.p. 86.0–86.5 °C; 76.5% yield; ^1^H NMR (400 MHz, Chloroform-*d*) *δ* 9.00 (m, 1H), 8.57 (m, 1H), 7.69 (s, 1H), 7.58 (m, 1H), 6.94–6.83 (m, 1H), 6.43 (d, *J* = 5.2 Hz, 1H), 4.78 (d, *J* = 11.0 Hz, 1H), 4.46 (m, 1H), 3.93 (d, *J* = 11.6 Hz, 1H), 3.43 (m, 1H), 3.17 (d, *J* = 11.6 Hz, 1H), 2.76 (m, 1H), 2.52 (m, 1H), 1.93 (d, *J* = 5.4 Hz, 1H), 1.92–1.84 (m, 1H), 1.75–1.66 (m, 1H), 1.49 (m, 2H), 1.42 (s, 1H), 1.40 (d, *J* = 3.2 Hz, 6H), 1.36 (s, 3H), 1.17 (d, *J* = 5.0 Hz, 1H), 1.13 (s, 3H), 1.10 (s, 3H), 1.08–1.02 (m, 1H), 0.95 (m, 1H). ^13^C NMR (101 MHz, Chloroform-*d*) *δ* 170.3, 151.0, 150.7, 147.5, 143.4, 136.4, 133.8, 128.7, 128.0, 127.9, 127.0, 126.4, 124.8, 122.3, 99.4, 76.9, 75.5, 71.8, 64.0, 48.1, 37.8, 37.6, 33.6, 31.7, 29.1, 26.4, 25.8, 25.4, 24.5, 22.3, 19.4, 18.3. HRMS (ESI) *m*/*z* 586.2122 [M + H]^+^, calculated for C_32_H_38_O_5_NCl_2_, 586.2122.

(14β)-(5′,7′-Dichloro-8′-quinolinoxy)-9-dehydro-17-hydro-3,19-isopropylideneoxy-andrographolide (**30**): white solid; m.p. 167.1–167.4 °C; 74.2% yield; ^1^H NMR (400 MHz, Chloroform-*d*) *δ* 8.98 (m, 1H), 8.57 (m, 1H), 7.69 (s, 1H), 7.58 (m, 1H), 6.87 (m, 1H), 6.44 (d, *J* = 5.0 Hz, 1H), 4.79 (d, *J* = 11.0 Hz, 1H), 4.48 (m, 1H), 3.93 (d, *J* = 11.6 Hz, 1H), 3.72 (m, 1H), 3.45 (m, 1H), 3.17 (d, *J* = 11.5 Hz, 1H), 2.77 (m, 1H), 2.49 (m, 1H), 1.96 (m, 2H), 1.93–1.86 (m, 1H), 1.71 (m, 1H), 1.46 (m, 1H), 1.40 (s, 3H), 1.39 (s, 3H), 1.36 (s, 3H), 1.33–1.28 (m, 1H), 1.20 (m, 2H), 1.16 (s, 3H), 0.97 (s, 3H). ^13^C NMR (101 MHz, Chloroform-*d*) δ 170.3, 150.6, 150.5, 147.6, 143.4, 136.3, 133.7, 128.9, 128.0, 127.9, 127.0, 126.3, 124.9, 122.3, 99.2, 76.9, 76.0, 71.9, 63.8, 48.5, 37.7, 37.6, 33.6 32.3, 28.8, 26.8, 25.9, 25.5, 24.8, 22.0, 19.4, 18.3. HRMS (ESI) *m*/*z* 586.2125 [M + H]^+^, calculated for C_32_H_38_NO_5_Cl_2_, 586.2122.

(14α)-(2′-Methyl-5′,7′-Dichloro-8′-quinolinoxy)-9-dehydro-17-hydro-3,19-isopropylideneoxy-andrographolide (**31**): white solid; m.p. 82.3–82.8 °C; 73.4% yield; ^1^H NMR (400 MHz, Chloroform-*d*) *δ* 8.44 (d, *J* = 8.7 Hz, 1H), 7.62 (s, 1H), 7.44 (d, *J* = 8.7 Hz, 1H), 6.88 (m, 1H), 6.41 (d, *J* = 5.0 Hz, 1H), 4.83 (m, 1H), 4.51 (m, 1H), 3.96 (d, *J* = 11.6 Hz, 1H), 3.47 (m, 1H), 3.20 (d, *J* = 11.5 Hz, 1H), 2.85 (m, 1H), 2.80 (s, 3H), 2.63 (m, 1H), 2.06–1.87 (m, 3H), 1.72 (m, 1H), 1.48 (m, 1H), 1.42 (s, 6H), 1.38 (s, 3H), 1.36–1.32 (m, 1H), 1.25–1.22 (m, 1H), 1.20 (d, *J* = 1.8 Hz, 1H), 1.18 (s, 3H), 1.03 (s, 3H), 0.88 (d, *J* = 3.8 Hz, 1H). ^13^C NMR (101 MHz, Chloroform-*d*) *δ* 170.3, 160.1, 150.3, 147.1, 142.9, 136.4, 133.7, 128.9, 127.7, 127.0, 126.8, 125.1, 124.6, 123.2, 99.2, 76.8, 76.0, 72.0, 63.8, 48.5, 37.7, 37.6, 33.6, 32.2, 28.8, 26.8, 25.9, 25.5, 25.3, 24.8, 22.0, 19.4, 18.3. HRMS (ESI) *m*/*z* 600.2281 [M + H]^+^, calculated for C_33_H_40_NO_5_Cl_2_, 600.2278.

(14β)-(2′-Methyl-5′,7′-dichloro-8′-quinolinoxy)-9-dehydro-17-hydro-3,19-isopropylideneoxy-andrographolide (**32**): white solid; m.p. 151.3–152.1 °C; 72.4% yield; ^1^H NMR (400 MHz, DMSO-*d*6) *δ* 8.90 (s, 1H), 8.49 (d, *J* = 8.7 Hz, 1H), 7.95 (s, 1H), 7.69 (d, *J* = 8.7 Hz, 1H), 6.53 (m, 1H), 6.46 (d, *J* = 4.5 Hz, 1H), 4.77 (m, 2H), 4.67 (d, *J* = 11.0 Hz, 1H), 4.58 (m, 1H), 3.83 (d, *J* = 11.6 Hz, 1H), 3.09 (d, *J* = 11.5 Hz, 1H), 2.75 (s, 3H), 1.33 (d, *J* = 10.3 Hz, 6H), 1.25 (s, 3H), 1.19 (d, *J* = 6.3 Hz, 9H), 1.08 (s, 3H), 0.85 (s, 3H). ^13^C NMR (101 MHz, Chloroform-*d*) *δ* 170.5, 160.2, 150.5, 147.2, 143.0, 136.5, 133.8, 129.0, 127.8, 127.1, 127.0, 125.2, 124.8, 123.3, 99.3, 76.2, 72.2, 70.2, 63.9, 48.7, 37.8, 37.7, 33.7, 32.4, 29.0, 27.0, 26.1, 25.6, 25.5, 25.0, 22.1, 19.5, 18.4. HRMS (ESI) *m*/*z* 600.2281 [M + H]^+^, calculated for C_33_H_40_NO_5_Cl_2_, 600.2278.

#### 4.3.5. Synthesis of Compounds **33** to **38**

Titled 9-dehydro-17-hydro series compounds **33** to **38** were prepared as shown in Figure 2 according to references [32,33,34]. A measure of 0.5 mmol of compounds **27** to **32** was dissolved in 4 mL of methanol and then treated with 0.05 mmol of TsOH·H_2_O at 20 °C for 30 min. The mixture was then diluted with ethyl acetate and washed with sat. aqueous NaHCO_3_ solution, brine, dried over anhydrous Na_2_SO_4_, filtered, and evaporated by a Rotavapor to dryness. Resulting compounds **33** to **38** were purified by silica gel column chromatography.

(14α)-(2′-Methyl-8′-quinolinoxy)-9-dehydro-17-hydro andrographolide (**33**): white solid; m.p. 148.3–149.1 °C; 68.5% yield; ^1^H NMR (400 MHz, Chloroform-*d*) *δ* 8.09 (d, *J* = 8.5 Hz, 1H), 7.55 (m, 1H), 7.42 (t, *J* = 7.8 Hz, 1H), 7.35 (d, *J* = 8.4 Hz, 1H), 7.14 (m, 1H), 6.96 (m, 1H), 6.21 (d, *J* = 5.4 Hz, 1H), 4.97 (s, 1H), 4.74 (m, 1H), 4.64 (m, 1H), 4.20 (m, 1H), 3.47–3.37 (m, 1H), 3.30 (d, *J* = 11.6 Hz, 1H), 3.04 (m, 1H), 2.77 (s, 3H), 2.52 (s, 1H), 2.03 (m, 3H), 1.75 (s, 4H), 1.56 (s, 1H), 1.55 (s, 3H), 1.23 (s, 3H), 0.94 (s, 2H), 0.80 (s, 3H). HRMS (ESI) *m*/*z* 492.2746 [M + H]^+^, calculated for C_30_H_38_NO_5_, 492.2744.

(14β)-(2′-Methyl-8′-quinolinoxy)-9-dehydro-17-hydro andrographolide (**34**): white solid; m.p. 143.3–143.9 °C; 73.2% yield; ^1^H NMR (400 MHz, Chloroform-*d*) *δ* 8.06 (d, *J* = 8.4 Hz, 1H), 7.59–7.47 (m, 1H), 7.39 (t, *J* = 7.9 Hz, 1H), 7.33 (d, *J* = 8.5 Hz, 1H), 7.12 (m, 1H), 6.93 (t, *J* = 6.4 Hz, 1H), 6.15 (d, *J* = 5.9 Hz, 1H), 4.95 (s, 1H), 4.72 (m, 1H), 4.61 (m, 1H), 4.18 (m, 1H), 3.44 (t, *J* = 8.1 Hz, 1H), 3.27 (s, 1H), 2.92 (d, *J* = 8.2 Hz, 2H), 2.74 (s, 3H), 2.39 (s, 1H), 2.01 (m, 2H), 1.77 (m, 2H), 1.72 (d, *J* = 14.5 Hz, 2H), 1.49 (s, 1H), 1.44 (s, 3H), 1.23 (s, 3H), 1.22–1.20 (m, 2H), 0.75 (s, 3H). ^13^C NMR (101 MHz, Chloroform-*d*) *δ* 170.2, 158.3, 151.9, 150.7, 141.1, 136.4, 136.1, 129.5, 128.2, 125.6, 125.0, 123.0, 122.6, 117.8, 80.5, 74.9, 71.7, 64.1, 51.6, 42.7, 38.4, 34.7, 34.2, 28.6, 28.0, 25.6, 22.5, 20.4, 19.4, 18.7. HRMS (ESI) *m*/*z* 492.2748 [M + H]^+^, calculated for C_30_H_38_NO_5_, 492.2744.

(14α)-(5′,7′-Dichloro-8′-quinolinoxy)-9-dehydro-17-hydro andrographolide (**35**): white solid; m.p. 88.7–89.2 °C; 79.2% yield; ^1^H NMR (400 MHz, Chloroform-*d*) *δ* 8.99 (m, 1H), 8.57 (m, 1H), 7.69 (s, 1H), 7.59 (m, 1H), 6.87 (m, 1H), 6.48 (d, *J* = 5.2 Hz, 1H), 4.78–4.72 (m, 1H), 4.43 (m, 1H), 4.17 (d, *J* = 11.2 Hz, 1H), 3.38 (m, 1H), 3.29 (d, *J* = 11.2 Hz, 1H), 2.79 (m, 1H), 2.70–2.50 (m, 2H), 2.42 (s, 1H), 1.97 (d, *J* = 6.6 Hz, 1H), 1.80–1.73 (m, 1H), 1.70 (d, *J* = 4.9 Hz, 1H), 1.54 (s, 1H), 1.44 (s, 3H), 1.21 (s, 3H), 1.13 (d, *J* = 12.6 Hz, 1H), 1.04–0.99 (m, 1H), 0.96 (d, *J* = 6.7 Hz, 1H), 0.90–0.87 (m, 1H), 0.85–0.82 (m, 1H), 0.79 (s, 3H). ^13^C NMR (101 MHz, Chloroform-*d*) *δ* 170.3, 150.9, 150.6, 147.5, 143.4, 135.6, 133.8, 129.4, 128.0, 127.9, 127.1, 126.4, 124.7, 122.3, 80.3, 76.8, 71.7, 64.1, 51.5, 42.7, 38.4, 34.6, 34.1, 28.6, 28.0, 22.4, 20.6, 19.4, 18.7. HRMS (ESI) *m*/*z* 546.1812 [M + H]^+^, calculated for C_29_H_34_NO_5_Cl_2_, 546.1809.

(14β)-(5′,7′-Dichloro-8′-quinolinoxy)-9-dehydro-17-hydro andrographolide (**36**): white solid; m.p. 139.8–140.6 °C; 73% yield; ^1^H NMR (400 MHz, Chloroform-*d*) *δ* 8.97 (m, 1H), 8.57 (m, 1H), 7.68 (s, 1H), 7.58 (m, 1H), 6.87–6.78 (m, 1H), 6.45 (d, *J* = 5.1 Hz, 1H), 4.77 (d, *J* = 11.1 Hz, 1H), 4.46 (m, 1H), 4.13 (d, *J* = 11.2 Hz, 1H), 3.49–3.35 (m, 1H), 3.25 (d, *J* = 11.1 Hz, 1H), 2.80 (m, 1H), 2.69 (s, 1H), 2.46 (m, 1H), 2.36 (s, 1H), 1.96 (d, *J* = 6.5 Hz, 2H), 1.80–1.67 (m, 3H), 1.53–1.46 (m, 1H), 1.40 (s, 3H), 1.30 (s, 1H), 1.21 (s, 3H), 1.15 (d, *J* = 12.5 Hz, 2H), 0.67 (s, 3H). ^13^C NMR (101 MHz, Chloroform-*d*) *δ* 170.3, 150.6, 150.1, 147.5, 143.3, 135.7, 133.8, 129.7, 128.0, 127.9, 127.0, 126.4, 125.0, 122.3, 80.4, 76.9, 71.9, 64.1, 51.6, 42.7, 38.3, 34.7, 34.1, 28.5, 27.9, 22.4, 20.4, 19.3, 18.6. HRMS (ESI) *m*/*z* 546.1812 [M + H]^+^, calculated for C_29_H_34_NO_5_Cl_2_, 546.1809.

(14α)-(2′-Methyl-5′,7′-dichloro-8′-quinolinoxy)-9-dehydro-17-hydro andrographolide (**37**): white solid; m.p. 115.1–115.7 °C; 76.3% yield; ^1^H NMR (400 MHz, DMSO-*d*6) *δ* 8.48 (d, *J* = 8.6 Hz, 1H), 7.93 (s, 1H), 7.68 (d, *J* = 8.6 Hz, 1H), 6.51 (m, 2H), 4.93 (s, 1H), 4.63 (d, *J* = 11.1 Hz, 1H), 4.54 (m, 1H), 4.01 (s, 1H), 3.76 (d, *J* = 10.9 Hz, 1H), 3.22 (d, *J* = 11.1 Hz, 1H), 3.10 (t, *J* = 8.2 Hz, 1H), 2.75 (s, 3H), 2.40 (m, 1H), 1.87 (d, *J* = 6.3 Hz, 2H), 1.66–1.49 (m, 3H), 1.39 (s, 3H), 1.33 (s, 3H), 1.02 (d, *J* = 7.8 Hz, 3H), 0.86 (s, 1H), 0.68 (s, 3H). ^13^C NMR (101 MHz, DMSO-*d*6) *δ* 169.7, 160.4, 148.7, 146.6, 142.5, 136.1, 133.4, 128.5, 127.0, 126.6, 126.2, 125.3, 124.2, 124.1, 78.1, 76.6, 71.5, 62.7, 51.2, 42.1, 38.1, 34.3, 33.9, 27.8, 27.6, 24.9, 22.9, 19.8, 19.1, 18.9. HRMS (ESI) *m*/*z* 560.1967 [M + H]^+^, calculated for C_30_H_36_NO_5_Cl_2_, 560.1965.

(14β)-(2′-Methyl-5′,7′-dichloro-8′-quinolinoxy)-9-dehydro-17-hydro andrographolide (**38**): white solid; m.p. 139.8–140.6 °C; 78.6% yield; ^1^H NMR (400 MHz, Chloroform-*d*) *δ* 8.42 (d, *J* = 8.7 Hz, 1H), 7.60 (s, 1H), 7.42 (d, *J* = 8.7 Hz, 1H), 6.81 (m, 1H), 6.39 (d, *J* = 5.0 Hz, 1H), 4.79 (d, *J* = 11.0 Hz, 1H), 4.46 (m, 1H), 4.14 (d, *J* = 11.2 Hz, 1H), 3.42 (m, 1H), 3.26 (d, *J* = 11.2 Hz, 1H), 2.86 (m, 1H), 2.77 (s, 3H), 2.71 (s, 1H), 2.58 (m, 1H), 2.34 (s, 1H), 2.05–1.91 (m, 2H), 1.74 (m, 3H), 1.49 (m, 1H), 1.39 (s, 3H), 1.33 (d, *J* = 12.9 Hz, 1H), 1.22 (s, 3H), 1.20–1.13 (m, 2H), 0.71 (s, 3H). ^13^C NMR (101 MHz, Chloroform-*d*) *δ* 170.4, 160.1, 150.0, 147.0, 142.9, 135.8, 133.7, 129.7, 127.7, 126.9, 126.9, 125.1 124.7, 123.2, 80.4, 72.0, 64.1, 51.6, 42.7, 38.3, 34.6, 34.2, 31.6, 28.6, 27.9, 25.3, 22.4, 20.3, 19.3, 18.7. HRMS (ESI) *m*/*z* 560.1967 [M + H]^+^, calculated for C_30_H_36_NO_5_Cl_2_, 560.1965.

### 4.4. Cell Lines and Treatments

Human HEK293 cells (ATCC, # CRL-1573) were cultured at 37 °C, 5% CO_2_ in DMEM supplemented with 10% FBS, penicillin (100 U/mL), and streptomycin (50 mg/mL). Human SH-SY5Y neuroblastoma cells (gift from Dr Narisorn Kitiyanant, Mahidol University) were grown at 37 °C, 5% CO_2_ in high glucose-DMEM supplemented with 10% FBS, penicillin (100 U/mL), and streptomycin (50 mg/mL). Andrographolide was prepared as a 100 mM stock solution (in 100% DMSO) and derivatives were prepared as a 10 mM stock solution (in 100% DMSO) from which serial dilutions were prepared. Cells were treated for 24 h before being processed for sAPPα secretion and Western blot analysis. For all conditions (including non-treated controls), DMSO was adjusted to 0.1%.

### 4.5. sAPPα Secretion and Detection

Secretion and detection of sAPPα in HEK293 and SH-SY5Y cells with the human-specific monoclonal anti-sAPPα antibody (2B3) has been previously described [58]. Briefly, following treatments in complete media, media was removed, and cells were incubated with fresh DMEM (1 mL) and allowed to secrete for 5 h. Then, 10% TCA precipitation of the whole medium was performed, and the precipitate was subjected to electrophoresis through 10% SDS-PAGE gels, transferred onto nitrocellulose membranes (100 min, 90 volts), incubated in 5% non-fat milk blocking solution for 30 min and incubated overnight at 4 °C with 2B3 (1 μg/mL). After three washes with PBST (PBS containing 0.05% Tween 20), membranes were then incubated with a HRP-conjugated anti-mouse IgG antibody (dilution 1/3000), rinsed three times with PBST incubated with ECL reagent, and signals were detected using an Azure c400 (Azure Biosystems, Dublin, CA, USA). Band densities were measured with the Image J software.

### 4.6. Western Blot Analyses

Cells were collected with phosphate-buffered saline (PBS)-EDTA and resuspended in 70 to 100 μL of lysis buffer (10 mM Tris/HCl, pH 7.5, 150 mM NaCl, 0.5% triton X-100, 0.5% deoxycholate, 5 mM EDTA). Protein concentrations were determined by the Bradford method [59] and 20–40 μg proteins were loaded onto 10% of SDS-PAGE gels which were run at 100 volts for 2–2.5 h. Proteins were then transferred onto nitrocellulose membranes for 60–120 min at 90 V). Protein transfer was verified by Ponceau red staining, and nitrocellulose membranes were subsequently incubated in 5% non-fat milk blocking solution for 45 min. Membranes were then incubated with primary antibodies directed against βAPP (dilution 1/2000), ADAM10 (dilution 1/500), BACE1 (dilution 1/1000), or β-actin (dilution 1/5000) on a platform shaker overnight at 4 °C. Bound antibodies were detected using goat anti-mouse (dilution 1/3000, polyclonal 7076, Cell Signaling) or goat anti-rabbit peroxidase-conjugated antibody (dilution 1/3000, polyclonal 7074, Cell Signaling). After 3 washes with PBST, membranes were incubated with a HRP-conjugated anti-rabbit (ADAM10, βAPP and BACE1) or anti-mouse (β-actin) secondary antibody (1/3000) for 2 h, rinsed 3 times with PBST, and processed as described above. All protein levels were normalized using β-actin as an internal standard.

### 4.7. Cell Viability Assay

Cells were seeded in 96-well polystyrene-coated tissue culture plates (Corning) over night. Proper attachment and confluence of the cells were confirmed by checking under a microscope. Media was removed and cells were treated with the compounds at various concentrations in quadruplicate for 24 h with control cells being treated with vehicle (0.1% DMSO). Media was then removed and cells were washed with sterile milli-Q water. Then, (3-(4,5-dimethylthiazol-2-yl)-2,5-diphenyltetrazolium bromide (MTT) was added (0.5 mg/mL) for 1.5 h. Formazan crystals formation was then checked under a microscope and the media was replaced by 100% DMSO until the purple crystals dissolved. Absorbance was measured at 595 nm.

### 4.8. Real-Time Quantitative Polymerase Chain Reaction (q-PCR)

Following treatments without (control) or with andrographolide or andrographolide derivative for 24 h at 37 °C in 1 mL of DMEM containing 1% FBS, total RNA was extracted from HEK293 or SH-SY5Y cells and purified with the PureLink RNA mini kit (Ambion, Life Technologies, Austin, TX, USA). Real-time PCR was performed with 100 ng of total RNA using the QuantiFast SYBR Green RT-PCR kit (Qiagen, Singapore) detector system (Eppendorf Mastercycler ep RealPlex, Eppendorf, Hamburg, Germany) and the SYBR Green detection protocol. The 2x QuantiFast SYBR Green RT-PCR master mix, QuatiFast RT mix, QuantiTectPrimer Assay, and template RNA were mixed and the reaction volume was adjusted to 25 μL using RNase-free water. The specific primers were designed and purchased from Qiagen. Each primer was added to a 10× QuantiTect Primer Assay containing a mix of forward and reverse primers for specific targets: Hs_ADAM10_1_SG (QT00032641, human ADAM10), Hs_BACE1_1_SG (QT00084777, human BACE1), and Hs_GAPDH_1_SG (QT00079247, human GAPDH, housekeeping gene for normalization).

### 4.9. α-Secretase Fluorimetric Assay on Intact Cells

SH-SY5Y and HEK293 cells were cultured in 35 mm-dishes coated with polylysine (10 μg/mL) until cells reached 80% confluence. Cells were treated in duplicate without (control) or with various concentrations of andrographolide or andrographolide derivatives for 24 h at 37 °C in 1 mL of DMEM containing 1% FBS. Duplicates were then incubated for 30 min at 37 °C in the absence or in the presence of the general metalloprotease inhibitor *o*-phenanthroline (100 μM) in 1.5 mL of PBS. Then, the α-secretase-specific JMV2770 substrate (10 μM) [60] was directly added into the media and cells were maintained at 37 °C. Every 15 min, 100 μL of media were removed and the α-secretase-specific activity corresponding to the *o*-phenanthroline-sensitive fluorescence was recorded in black 96-well plates at 320 nm and 420 nm excitation and emission wavelengths, respectively.

### 4.10. β-Secretase Fluorimetric Assay on Cell Homogenates

SH-SY5Y and HEK293 cells were cultured in 35 mm-dishes until they reach 80% confluence, treated without (control) or with various concentrations of andrographolide or andrographolide derivatives for 24 h at 37 °C in DMEM containing 1% FBS and assayed for their β-secretase activity as previously described [61]. Briefly, cells were collected, lysed with Tris 10 mM pH 7.5, homogenized, and kept on ice. Samples were assayed for their protein contents with the Bradford method and adjusted to a 3 μg/μL concentration. Then, 30 μg of each sample (10 μL) diluted in 10 mM sodium acetate buffer pH 4.5 were incubated for 30 min at 37 °C in black 96-well plates (in a final volume of 100 μL) in the absence (triplicate) or in the presence (triplicate) of the β-secretase specific inhibitor JMV1197. Then, the β-secretase-specific JMV2236 substrate (10 μM) was added to all samples and plates were maintained at 37 °C. Every 15 min, the β-secretase-specific activity corresponding to the JMV1197-sensitive fluorescence was recorded at 320 nm and 420 nm excitation and emission wavelengths, respectively.

### 4.11. Statistical Analysis

Statistical analyses were performed with the Prism software (GraphPad, San Diego, USA) using an unpaired *t*-test for pairwise comparisons. All results are expressed as means ± SEM and *p* values equal to or less than 0.05 were considered significant.

## Data Availability

The data presented in this study are available in this article.

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
