# Peer review of "Synthesis and Characterization of Andrographolide Derivatives as Regulators of βAPP Processing in Human Cells"

_molecules, 2021, doi:10.3390/molecules26247660_

Round 1

Reviewer 1 Report

Authors claimed that some of andrographolide-based compounds have activating effect on a-secretase and inhibiting effect on b-secretase. These results may be useful for therapeutic treatment of AD in the future.

Based on the fact that andrographolide shifts APP metabolism towards the non-amyloidogenic pathway, the authors initially tested whether the synthesized andrographolide derivatives act as a-secretase activators. Some compounds indeed increased mRNA level of ADAM10, and increased products from fluorimetric substrate. However, based on their findings, they acted mostly as b-secretase inhibitors. Since a-, b-secretases may compete for the same APP, increased level of sAPPa by these compounds could be via inhibiting b-secretase.

Major concern:

  1. The manuscript (Introduction and Result sections) should be re-organized to emphasize that andrographolide-based compounds act mostly as b-secretase inhibitors not a-secretase activators.

At the same token, SAR in Table 1 is not needed because it is based on the assumption that the compounds are a-secretase activators.

  1. The effects of these compounds on the production of Ab40 and Ab42 should be included.
  2. In order to show that andrographolide-based compounds act as b-secretase inhibitors competing for the same APP, levels of sAPPb should be measured in addition to levels of sAPPa.

Minor concern:

  1. sAPPa. western blot in Fig. 2A and Fig. 4B did not match with the corresponding bar graphs. They should be replaced with more representative results.

Reviewer 2 Report

The manuscript describes the synthesis and biological evaluation of a novel series of andrographolide-based derivatives series acting on the metabolism of bAPP processing. While andrographolide and its analog 9 positively activated α-secretase, the 17-hydro-9-dehydro compounds 31 and 37 were found as β-secretase inhibitors. The two mechanisms of action are considered beneficial for the treatment of Alzheimer's disease. The manuscript is well described, the chemistry looks solid and the methodology is robust.

However, some aspects may be improved as follows:

In the abstract, in place of compounds numbering code it would be better to report the chemical name or, as better, the main chemical features to describe them.

Add to reference 2, the more recent citation:

Campora M. et al. Journey on Naphthoquinone and Anthraquinone Derivatives: New Insights in Alzheimer's Disease. Pharmaceuticals 2021, 14, 33;

Line 88: the compound numbering code should be in bold style. Please check and revise all the manuscript, including the legend of figures (f.i. Figure 4-6)

I suggest removing Table 1 from the manuscript, the reported information is not particularly informative.

The SAR analysis (lines 138-183) must be simplified; its actual form is very difficult to read with respect to the other sections of the paper that are well organized.

Round 2

Reviewer 1 Report

None.